# Spatial Heterogeneous of Ecological Vulnerability in Arid and Semi-Arid Area: A Case of the Ningxia Hui Autonomous Region, China

**Rong Li [1], Rui Han [1], Qianru Yu [1], Shuang Qi [2] and Luo Guo [1,*]**

[1] College of the Life and Environmental Science, Minzu University of China, Beijing 100081, China; 19301382@muc.edu.cn (R.L.); 18301234@muc.edu.cn (R.H.); 18301244@muc.edu.cn (Q.Y.)

[2] Department of Geography, National University of Singapore; Singapore 117570, Singapore; e0457619@u.nus.edu

[*] Correspondence: guoluo@muc.edu.cn

**Abstract:** Ecological vulnerability, as an important evaluation method reflecting regional ecological status and the degree of stability, is the key content in global change and sustainable development. Most studies mainly focus on changes of ecological vulnerability concerning the temporal trend, but rarely take arid and semi-arid areas into consideration to explore the spatial heterogeneity of the ecological vulnerability index (EVI) there. In this study, we selected the Ningxia Hui Autonomous Region on the Loess Plateau of China, a typical arid and semi-arid area, as a case to investigate the spatial heterogeneity of the EVI every five years, from 1990 to 2015. Based on remote sensing data, meteorological data, and economic statistical data, this study first evaluated the temporal-spatial change of ecological vulnerability in the study area by Geo-information Tupu. Further, we explored the spatial heterogeneity of the ecological vulnerability using Getis-Ord Gi*. Results show that: (1) the regions with high ecological vulnerability are mainly concentrated in the north of the study area, which has high levels of economic growth, while the regions with low ecological vulnerability are mainly distributed in the relatively poor regions in the south of the study area. (2) From 1990 to 2015, ecological vulnerability showed an increasing trend in the study area. Additionally, there is significant transformation between different grades of the EVI, where the area of transformation between a slight vulnerability level and a light vulnerability level accounts for 41.56% of the transformation area. (3) Hot-spot areas of the EVI are mainly concentrated in the north of the study area, and cold-spot areas are mainly concentrated in the center and south of the study area. Spatial heterogeneity of ecological vulnerability is significant in the central and southern areas but insignificant in the north of the study area. (4) The grassland area is the main driving factor of the change in ecological vulnerability, which is also affected by both arid and semi-arid climates and ecological projects. This study can provide theoretical references for sustainable development to present feasible suggestions on protection measures and management modes in arid and semi-arid areas.

**Keywords:** ecological vulnerability; geospatial analysis; sustainable development

## 1. Introduction

Ecological vulnerability, which refers to the self-recovery ability of ecosystems when they suffer disturbances at a specific temporal scale [1], has been an important concept for reflecting the deviation degree from the original ecological condition under the external interferences [2,3]. The assessment of ecological vulnerability, therefore, is critical for research on ecological change [4,5]. With the rapid development of society, however, the growth of anthropogenic activities has significantly aggravated ecological vulnerability on a global scale [6,7]. The sustainable development of arid and semi-arid

regions is related to assessing ecological vulnerability because arid and semi-arid areas are especially sensitive to variable environmental changes and human influences.

The assessment of ecological vulnerability is essential for ensuring and managing eco-environmental stability. Ecological vulnerability can be assessed by the analytic hierarchy process (AHP) [1,8,9], principal component analysis (PCA) [10,11], fuzzy comprehensive evaluation [12], entropy weight analysis [13,14], spatial principal component analysis (SPCA), and other techniques. Studies have shown that the SPCA method, which is based on principal component analysis (PCA) and spatial feature extraction, has advantages in ecological vulnerability assessment [1,15,16]. One advantage of this method is that the SPCA not only adds spatial constraints to the traditional PCA but also considers the spatial dependence in data sets. This method has been widely used to describe the characteristics of changes in ecological vulnerability in arid and semi-arid areas because of its advantages in being good for map and comparison services [17,18].

In terms of the recognition of ecosystem-humanity interactions, human activities are considered to have a serious impact on regional ecological vulnerability [19,20]. For example, previous research shows that population and economic growth lead to increased conditions of ecological vulnerability, especially in close proximity to urban agglomerations [21–23]. Other studies imply that ecological vulnerability is negatively correlated with urbanization [14]. Although these studies explain the relationship between ecological vulnerability and human activities, some gaps in knowledge remain. Most studies have focused on the establishment of indicator systems of ecological vulnerability [1] or the spatial and temporal changes in ecological vulnerability [4,24] but they have paid little attention to the spatial heterogeneity of ecological vulnerability [22]. Currently, ecological vulnerability assessment focuses on unstable ecosystems, such as urban areas [22,25], specific habitats [26], or areas with a poor ecological environment [19,27,28]. These ecological vulnerability analyses fail to explore the local spatial differentiation of ecological vulnerability. Getis-Ord Gi* is a spatial statistical method, which can describe and visualize the spatial distribution, discover local spatial correlation patterns, identify heterogeneous units, and suggest spatial states [29,30]. Compared with the traditional method of identifying process-related regions, Getis-Ord Gi* has shown a certain effectiveness in exploring the spatial heterogeneity of ecological vulnerability, which lays a foundation for the study of the impact of human interference on ecological vulnerability.

This study takes the Ningxia Hui Autonomous Region, at the intersection of the Loess Plateau, the Mongolian Plateau, and the Qinghai-Tibet Plateau in China, as a case to examine ecological vulnerability in arid and semi-arid regions. Over the past few decades, Ningxia has undergone rapid urbanization, large-scale agricultural modernization, and continuous ecological conservation projects. The poor ecological conditions and the various human influences make the study area an ideal region to study ecological vulnerability. The objectives of this study are to: (1) analyze changes of ecological vulnerability in Ningxia, (2) visualize the spatial-temporal transformation of ecological vulnerability in Ningxia based on Geo-information Tupu, (3) explore the spatial heterogeneity of ecological vulnerability based on Getis-Ord Gi*, and (4) detect the driving factors of ecological vulnerability by SPCA. This study provides references for sustainable development in arid and semi-arid areas.

## 2. Materials and Methods

### 2.1. Study Area

The Ningxia Hui Autonomous Region (Ningxia) is located in the upper and middle reaches of the Yellow River in the northwest of China (Figure 1). The study area is situated at 35°14′–39°23′ N and 104°17′–107°39′ E in an arid and semi-arid region, which belongs to the temperate continental climate. The annual average temperature is about 5–9C. The annual precipitation is 150–600 mm, and the average annual surface evaporation is 1250 mm. The study area consists primarily of plains covered with grassland and farmland, which have provided the main agricultural function of the study area since the 16th century. In 2015, the study area included 5 cities and 22 counties [31].

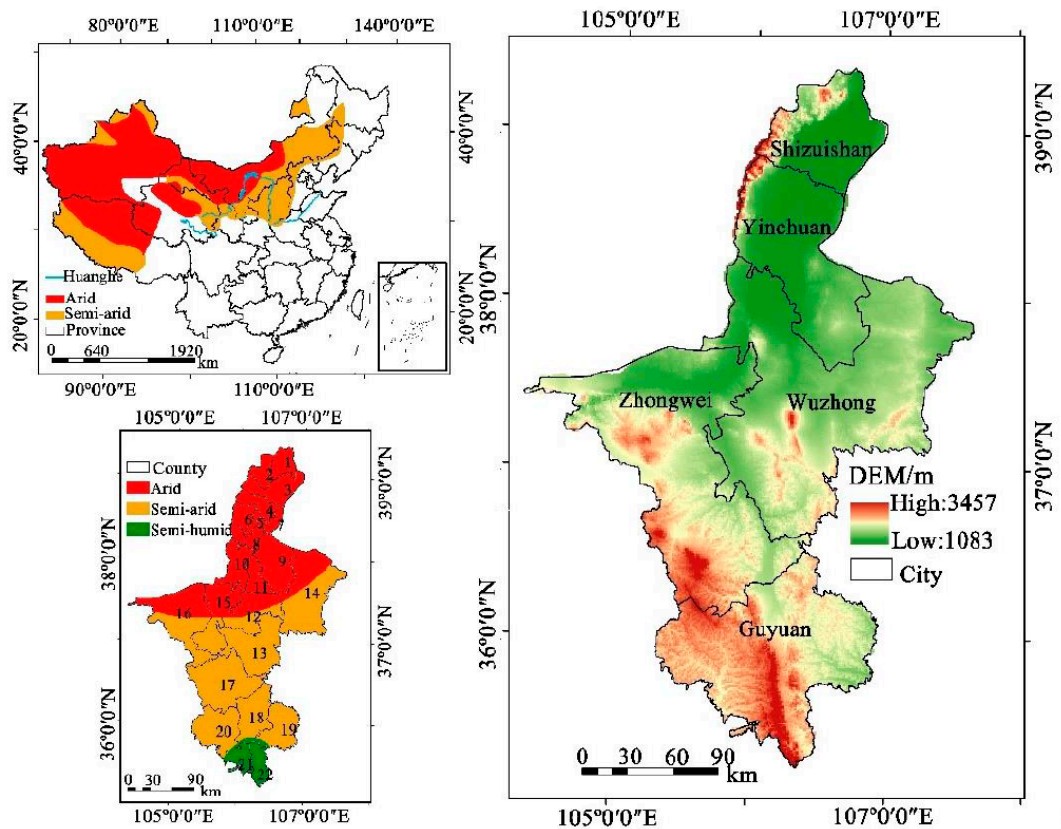

**Figure 1.** Location of the study area. (Numbers refer to counties: 1—Huinong district, 2—Dawukou district, 3—Pingluo county, 4—Helan county, 5—Jinfeng district, 6—Xixia district, 7—Xingqing district, 8—Yongning county, 9—Lingwu, 10—Qingtongxia, 11—Litong district, 12—HongSiBao district, 13—Tongxin county, 14—Yanchi county, 15—Zhongning county, 16—Shapotou district, 17—Haiyuan county, 18—Yuanzhou district, 19—Pengyang county, 20—Xiji county, 21—Longde county, and 22—Jingyuan county).

### 2.2. Data Collection

In this study, land use/land cover (LULC) data from 1990, 1995, 2000, 2005, 2010, and 2015 was provided by the Data Center for Resources and Environmental Sciences, Chinese Academy of Sciences (RESDC). The population density (POP) data was also from RESDC. The gross domestic product (GDP) data came from National Earth System Science Data Sharing Infrastructure, National Science and Technology Infrastructure of China. All spatial resolution was 1 km.

### 2.3. Methods

This study used the Data Management Tool module to generate cells by ArcGIS 10.4 and to perform all calculations for internal landscape units. This study adopted the sampling grids of 1 × 1 km with a total 66,400 cells for the six periods in this study area, which used ArcGIS 10.4, Python, and GeoDa 1.1 for further analyses.

#### 2.3.1. Assessment of Ecological Vulnerability

Given the existing international evaluation principles and standards [1,2], the comprehensive evaluation system of ecological vulnerability was established combining the ecological conditions of the study area. The ecological vulnerability index is based on both natural and social factors. The natural factors include the digital elevation model (DEM), hours of sunshine, average annual precipitation, average annual temperature, normalized difference vegetation index (NDVI), soil erosion, and degree of land use. Social factors include the gross domestic product (GDP), agricultural output, industrial output, population density, and grassland area. All indicators are standardized. SPCA is employed to

determine the weight of each factors. SPCA is used to add spatial features on the basis of PCA, and its calculation principle is consistent with PCA [1,15,16].

PCA is a statistical analysis method that transforms multiple variables into a few principal components (composite indicators) through dimensionality reduction [32–34]. In this study, PCA was used to make a linear combination of 12 standardized indexes to make them become new comprehensive indexes. The correlation coefficient matrix was solved to obtain the eigenvectors, thus obtaining 12 principal component results. The number of principal components was determined by the standard of a cumulative contribution rate ≥ 85%. From this w got our final principal component result. The calculation formula was as follows:

$$R = \frac{Z^T Z}{n} \tag{1}$$

$$|R - \lambda I| = 0 \tag{2}$$

$$CCR = \frac{\sum_{j=1}^{m} \lambda_j}{\sum_{j=1}^{n} \lambda_j} \geq 0.85 \tag{3}$$

$$P = Z \cdot W \tag{4}$$

where $R$ was the correlation coefficient matrix, $Z$ was the standardized value of each selected index, $n$ was the number of indexes, $\lambda$ was the eigenvalues of the $R$ correlation coefficient matrix, $I$ was the identity matrix, $CCR$ was the cumulative contribution rate, $m$ was the number of principal components that were determined, $P$ was the matrix containing values of every considered principal component, and $W$ was $m$ number of eigenvectors with the largest eigenvalues selected to form the dimensional matrix. The SPCA was obtained by calculating PCA in ArcGIS 10.4. The SPCA results are shown in Table 1.

**Table 1.** The results of spatial principal component analysis.

|  | Principal Component | 1990a | 1995a | 2000a | 2005a | 2010a | 2015a |
|---|---|---|---|---|---|---|---|
| **Eigenvalue/%** | I | 0.069 | 0.073 | 0.066 | 0.067 | 0.066 | 0.066 |
|  | II | 0.036 | 0.039 | 0.038 | 0.042 | 0.041 | 0.044 |
|  | III | 0.021 | 0.019 | 0.024 | 0.029 | 0.026 | 0.026 |
|  | IV | 0.015 | 0.013 | 0.015 | 0.016 | 0.016 | 0.017 |
| **Contribution/%** | I | 42.315 | 44.965 | 40.530 | 37.930 | 38.657 | 37.612 |
|  | II | 22.256 | 24.309 | 23.420 | 23.765 | 23.650 | 24.640 |
|  | III | 13.157 | 11.545 | 14.908 | 16.288 | 15.106 | 14.890 |
|  | IV | 9.453 | 7.748 | 9.526 | 9.006 | 9.108 | 9.543 |
| **Cumulative contribution/%** | I | 42.315 | 44.965 | 40.530 | 37.930 | 38.657 | 37.612 |
|  | II | 64.571 | 69.274 | 63.950 | 61.694 | 62.306 | 62.252 |
|  | III | 77.729 | 80.819 | 78.857 | 77.983 | 77.412 | 77.143 |
|  | IV | 87.182 | 88.566 | 88.384 | 86.988 | 86.520 | 86.686 |

Based on spatial principal component analysis (SPCA), the ecological vulnerability index (EVI) was the sum of the weighted principal components [2].

$$EVI = \sum_{i=1}^{m} r_m P_m \tag{5}$$

$$r_i = \frac{n_i}{\sum_i^m n_i} \tag{6}$$

where *EVI* was the ecological vulnerability index, *r* was the contribution ratio, *P* is the principal component, *m* was the number of principal components, *r* $_i$ was the contribution ratio of principal component *i*, and $n_i$ was the eigenvalue of principal component *i*.

### 2.3.2. Gradient Classification of Ecological Vulnerability

In this paper, natural breaks classification (NBC) was used to classify the EVI to reflect different degrees of ecological vulnerability. NBC is a method of analyzing the statistical distribution of attribute space, which maximizes the difference between classes. In this study, the assessment of ecological vulnerability was divided into five grades [4,24], namely slight vulnerability: <0.1444, light vulnerability: 0.1444–0.2480, medium vulnerability: 0.2480–3982, heavy vulnerability: 0.3982–0.6282, and extreme vulnerability: >0.6282.

### 2.3.3. Geo-Information Tupu Change Analysis

Geo-information Tupu is an effective way to study the process integration of ecological vulnerability. Based on the ArcGIS tool, EVI process change is revealed by Geo-information Tupu [35]. The specific operational formula was [36]

$$T = Y_1 \times 10^{n-1} + Y_2 \times 10^{n-2} + \ldots + Y_n \times 10^{n-n} \tag{7}$$

where T was the Tupu unit code of the Tupu mode characteristics within the representation research stage, $Y_n$ was the ecological vulnerability Tupu unit code of representation in a certain year, and n was the number of the ecological vulnerability grade.

### 2.3.4. Cold-Hotspot Study Change Analysis

In this study, the Getis-Ord Gi* index was used to analyze the high/low spatial aggregation degree of EVI changes, that is, the spatial distribution of cold/hot spots. The calculation formula was

$$Gi^* = \frac{\sum_{j=1}^{n} w_{ij} x_j - \overline{X} \sum_{j=1}^{n} w_{ij}}{s \sqrt{\left[ n \sum_{j=1}^{n} w_{ij}^2 - \left( \sum_{j=1}^{n} w_{ij} \right)^2 \right] / (n-1)}} \tag{8}$$

$$\overline{X} = \frac{1}{n} \sum_{j=1}^{n} x_j \tag{9}$$

$$\sqrt{\left( \frac{1}{n} \sum_{j=1}^{n} x_j^2 - \overline{X}^2 \right)} \tag{10}$$

where $G_i^*$ was the output statistical *Z*-score, $x_j$ was the EVI change of space unit *j*, and $w_{ij}$ was the spatial weight between adjacent space units *i* and *j*.

## 3. Results

### 3.1. Spatial and Temporal Characteristics of Ecological Vulnerability

The results of the ecological vulnerability index (EVI) in Ningxia were as follows:

1. $EVI_{1990} = 0.4852A_1 + 0.2553A_2 + 0.1509A_3 + 0.1084A_4$;
2. $EVI_{1995} = 0.5077B_1 + 0.2745B_2 + 0.1303B_3 + 0.0875B_4$;
3. $EVI_{2000} = 0.4586C_1 + 0.2650C_2 + 0.1687C_3 + 0.1078C_4$;
4. $EVI_{2005} = 0.4360D_1 + 0.2732D_2 + 0.1872D_3 + 0.1035D_4$;
5. $EVI_{2010} = 0.4468F_1 + 0.2733F_2 + 0.1746F_3 + 0.1053F_4$;

6. 　　$EVI_{2015} = 0.4339P_1 + 0.2842P_2 + 0.1718P_3 + 0.1101P_4;$

where $EVI_i$ were the ecological vulnerability index in year $i$; A1~A4, B1~B4, C1~C4, D1~D4, F1~F4, and P1~P4 were the principal components in 1990, 1995, 2000, 2005, 2010, and 2015, respectively.

Figure 2 shows the area percentages of different grades of ecological vulnerability. It can be seen that the sum of the area percentage (SSL) of slight vulnerability and light vulnerability of the whole area was ranked as $SSL_{2010} > SSL_{2005} > SSL_{2000} > SSL_{2015} > SSL_{1990} > SSL_{1995}$. In 2005, the whole degree of ecological vulnerability was at the lowest level, with the area of slight vulnerability accounting for 51% of the study area.

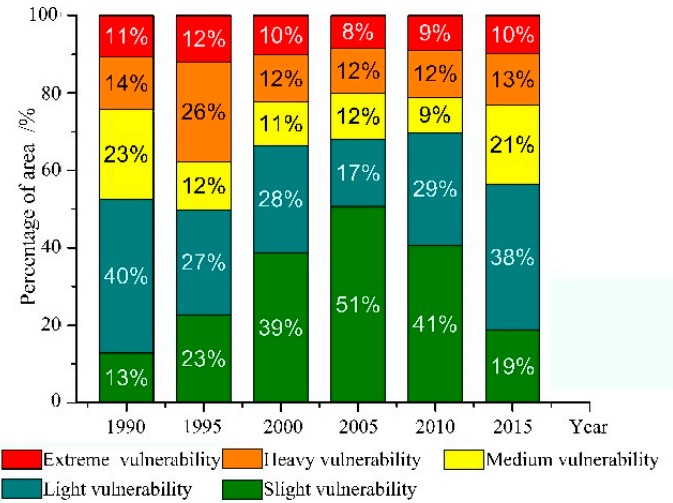

**Figure 2.** Changes of ecological vulnerability in Ningxia.

The spatial change of ecological vulnerability is shown in Figure 3. The spatial statistical model was used for data visualization and shows the spatial and temporal variation of ecological vulnerability in Ningxia more clearly and intuitively. The EVI showed a decreasing trend from north to south. The grade of ecological vulnerability in the north of Ningxia (Shizuishan City and Yinchuan City) from 1990 to 2015 was mainly extreme vulnerability, and the area of extreme vulnerability initially increased but decreased later. From 1990 to 2015, the grade of ecological vulnerability in the west of Ningxia (Zhongwei City) mainly consisted of medium vulnerability and heavy vulnerability. The areal proportions of medium vulnerability presented a trend of "increased-decreased-increased", while the area of heavy vulnerability initially increased but decreased later. The grade of ecological vulnerability in the east of Ningxia (Wuzhong City) was light vulnerability, and the area of light vulnerability initially increased but decreased in 2015. The grade of ecological vulnerability in the south of Ningxia (Guyuan City) mainly consisted of slight vulnerability and light vulnerability, and the area of slight vulnerability showed initially increased but decreased in the latter period.

*3.2. Transformation of Ecological Vulnerability*

Tupu is a spatial model that visualizes the transformation between different grades of ecological vulnerability at each patch. Tupu analysis can more clearly show the dynamic change of ecological vulnerability in Ningxia from 1990 to 2015 (Figure 4). In the past 25 years, the main changes of ecological vulnerability in the study area were that the area of light vulnerability decreased and the area of slight vulnerability increased. First, from 1990 to 2015, the transformation area of ecological vulnerability level accounted for 37.95% of the study area. There were 20 types of transformation, among which the main type was the transformation from light vulnerability to slight vulnerability, accounting for 10.37% the study area. Secondly, the area of slight vulnerability transforming into light vulnerability accounted for 5.39% of the study area.

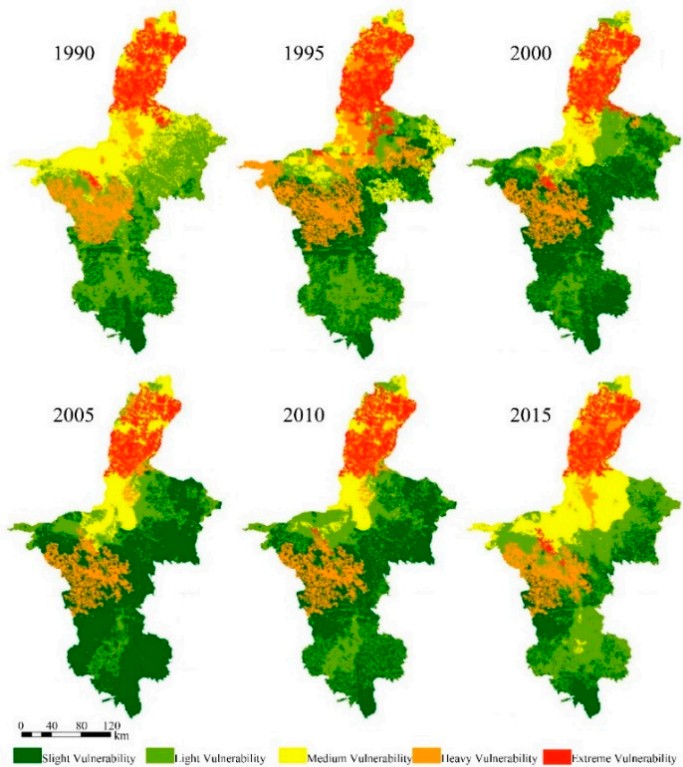

**Figure 3.** Spatial distribution of ecological vulnerability in the study area.

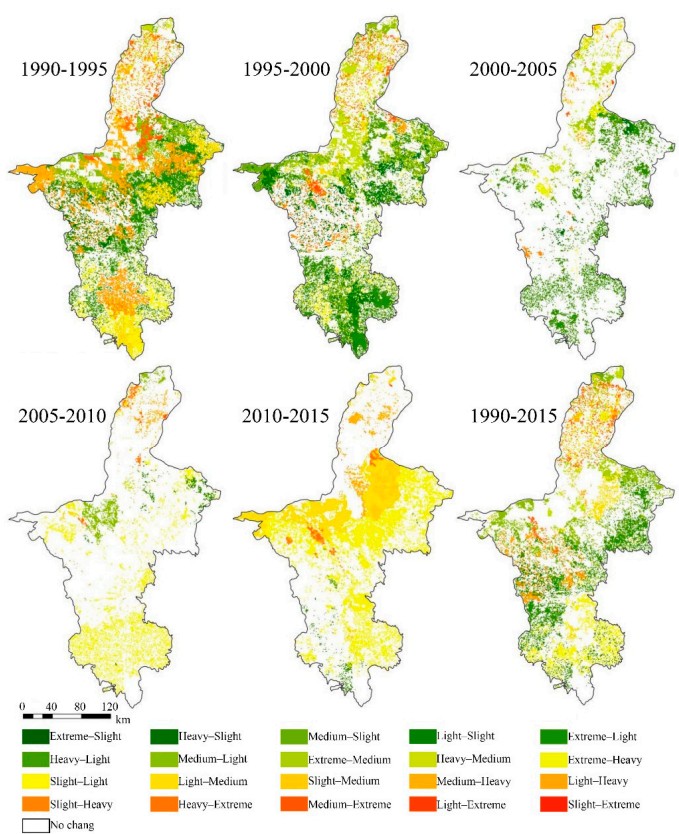

**Figure 4.** Tupu of study area ecological vulnerability.

　　There were 20 types of Tupu changes of ecological vulnerability, and the characteristics of Tupu changes were different in six periods. The transformation areas of ecological vulnerability accounted for 58.12%, 52.26%, 24.93%, 19.97%, 17.13%, and 38.43%, respectively, of the study area for the years of 1990, 1995, 2000, 2005, 2010, and 2015. The main change of ecological vulnerability was from light vulnerability to slight vulnerability, and the change was significant in the period of 1990–1995, followed by 1995–2000. Such changes mainly occurred in the center of the study area in 1990–1995 and in the center and south of the study area in 1995–2000. The transformation of ecological vulnerability was less from 2005 to 2010, when the main type was transformation from slight vulnerability to light vulnerability in the center of the study area.

### 3.3. Cold-Hotspot Analysis of EVI

　　Cold-hotspot is a spatial model used to display spatial aggregation calculated by Getis-Ord Gi*. The calculation of the EVI of Ningxia from 1990 to 2015 based on the cold-hotspot can intuitively see spatial aggregation of similar values. At the same time, by comparing the spatial distribution of cold spots and hot spots in different years, we can get the differences in the spatial change of EVI on the time scale (Figure 5). On the whole, the hot-spot areas were clustered in the north and the cold-spot areas were distributed in the south of the study area. In terms of the temporal trend, areas of hot spots and cold spots of the EVI all decreased from 1990 to 2005, but the trend was reversed from 2005 to 2015. This trend hints that north-south heterogeneity of the EVI was mitigated in the former period (1990–2005), while the changes of ecological vulnerability since 2005 are noteworthy.

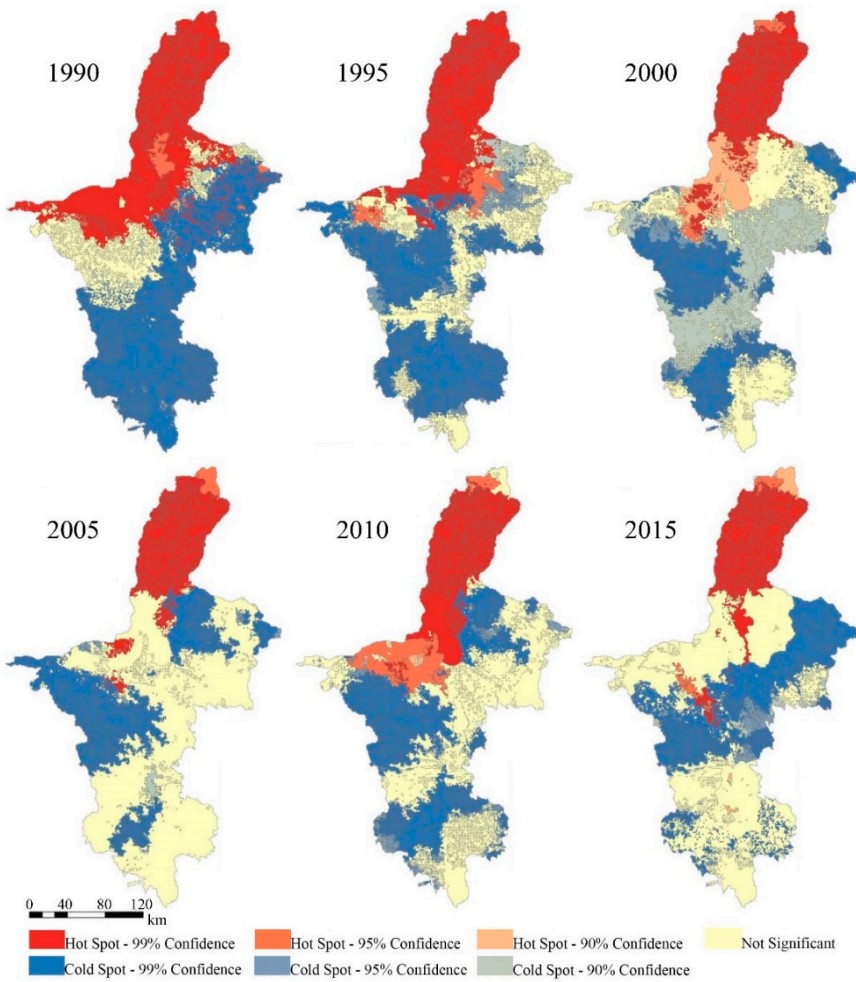

**Figure 5.** Getis-Ord Gi* scores of ecosystem vulnerability values.

In terms of spatial patterns, planners need to pay more attention to the high ecological vulnerability in the north and increasing ecological vulnerability in the south. In the north, the ecological conditions in Shizuishan City and Yinchuan City were always at a disadvantage from 1990 to 2015 (hot spots of the EVI). In the south, the cold spots of Guyuan City showed a decreasing trend, which means an increasing ecological sensitivity. Fortunately, environmental conditions in the west and east of the study area seemed to become better during the study period. In the west, hot spots of the EVI in Zhongwei City decreased, implying that the ecological vulnerability weakened. In the east, Wuzhong City was at a low level of ecological vulnerability in 2015, showing that the environmental condition was in a good state, although it seemed to suffer from 1990 to 2010.

### 3.4. Driving Factors Analysis

Principal component vectors were employed to detect the driving factors of ecological vulnerability change, as shown in Figure 6.

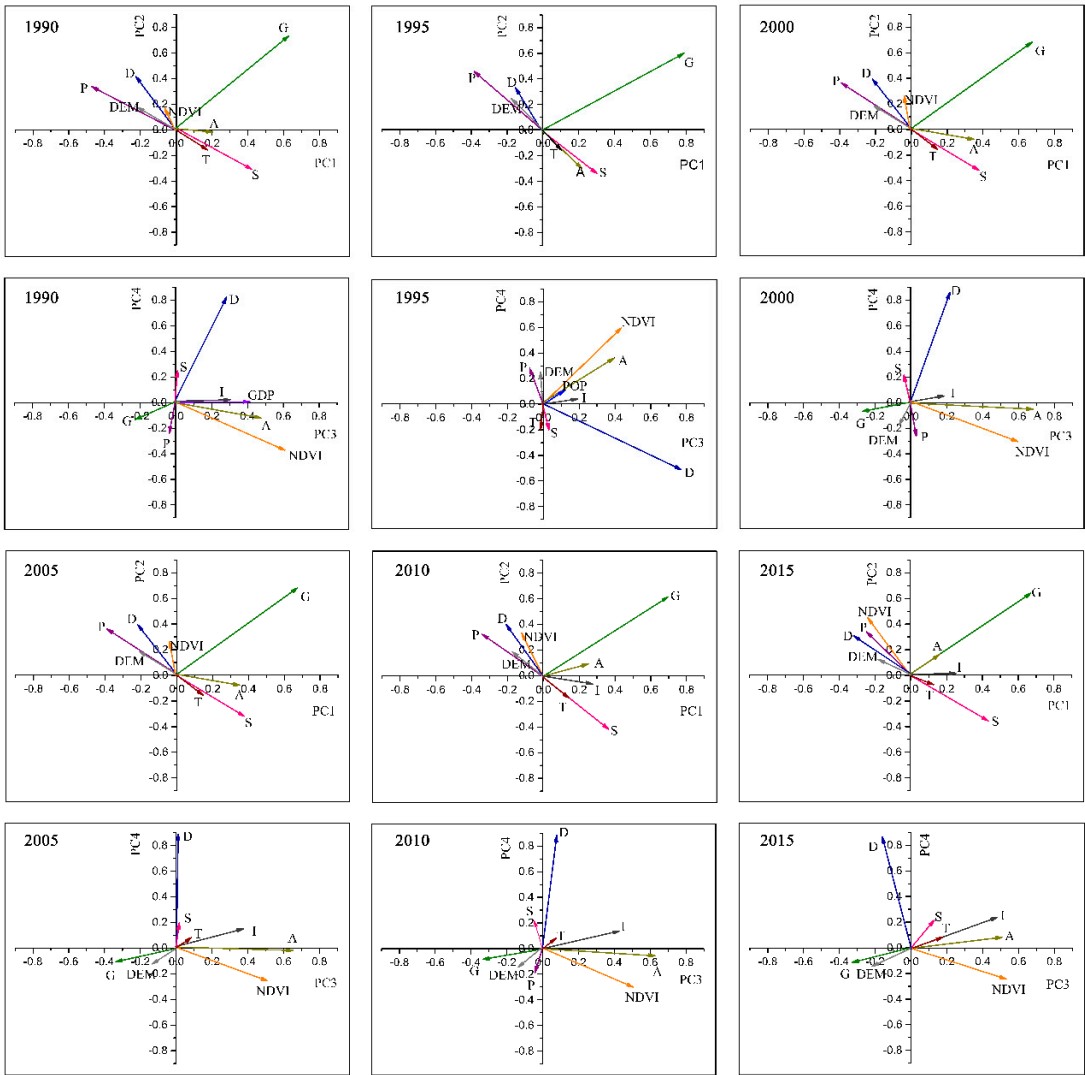

**Figure 6.** Correlation analysis diagram of principal components and factors. DEM—Digital Elevation Model, P—Annual Average Precipitation, T—Annual Average Temperature, S—Hours of Sunshine, NDVI—Normalized Difference Vegetation Index, SE—Soil Erosion, D—Degree of Land Use, A—Agricultural Output, I—Industrial Output, POP—Population Density, GDP—Gross Domestic Product, and G—Grassland Area.

In 1990, principal components I, II, III, and IV all reflected socio-economic background factors, while principal components I and II also reflected meteorological factors. The contribution rate of principal component I was 42.32%. This factor was highly correlated with the grassland area, hours of sunshine, and GDP, with the correlation coefficients of 62.87%, 41.53%, and 16.86%, respectively. The contribution rate of principal component II was 22.26%. This factor was highly correlated with the grassland area, degree of land use, and annual average precipitation, with the correlation coefficients of 73.23%, 42.13%, and 33.08%, respectively. The contribution rate of principal component III was 13.16%. The correlation coefficient between this factor and the NDVI, agricultural output, and GDP was high at 60.68%, 47.69%, and 41.14%, respectively. The contribution rate of principal component IV was 9.45%, and the correlation coefficient between this factor and the degree of land use was higher (82.53%). The GDP, agricultural output, and land use are commonly used indicators to describe factors of social and economic background. Therefore, principal components I, II, III, and IV have received a more comprehensive response in human society and economy. It can be seen that human social and economic factors were the main driving force of ecological vulnerability in Ningxia. At the same time, the hours of sunshine and annual average precipitation are the main indicators used to describe meteorological factors. Principal component I and principal component II were highly correlated with the hours of sunshine and annual average precipitation, respectively. It can be seen that meteorological factors were external forces affecting the ecological vulnerability of Ningxia.

In 1995, principal components II, III, and IV all reflected socio-economic background factors. The contribution rate of principal component I was 44.97%, which was mainly correlated with the grassland area (79.42%) and hours of sunshine (29.48%). The contribution rate of principal component II was 24.31%, which was mainly correlated with the grassland area (59.80%), agricultural output (46.22%), and degree of land use (33.40%). The contribution rate of principal component III was 11.55%, which was mainly correlated with the degree of land use (76.91%), NDVI (43.57%), and agricultural output (39.83%). The contribution rate of principal component IV was 7.75%, which was mainly correlated with the NDVI (59.75%) and agricultural output (36.47%). Agricultural output and degree of land use are common indicators to describe factors of social and economy background. Therefore, principal components II, III, and IV reflect that human social and economic factors are the main driving force of ecological vulnerability. The hours of sunshine are often the meteorological factor, so the meteorological factor was the external force that affected the ecological vulnerability of Ningxia.

In 2000, principal components I, II, III, and IV all reflected socio-economic background factors, while principal components I and II also reflected meteorological factors. The contribution rate of principal component I was 40.53%, which was mainly correlated with the grassland area, hours of sunshine, and agricultural output (67.19%, 37.58%, and 34.90%, respectively). The contribution rate of principal component II was 23.42%, which was mainly correlated with the grassland area, degree of land use, and annual average precipitation (68.32%, 39.90%, and 36.68%, respectively). The contribution rate of principal component III was 14.91%, which was mainly correlated with the agricultural output and NDVI (68.26% and 60.02%, respectively). The contribution rate of principal component IV was 9.53%, which was mainly correlated with the degree of land use (86.01%).

In 2005, principal components II, II, and IV all reflected socio-economic background factors. The contribution rate of principal component I was 37.93%, which was mainly correlated with the grassland area and hours of sunshine (66.34% and 42.09%, respectively). The contribution rate of principal component II was 23.77%, which was mainly correlated with the grassland area, degree of land use, and NDVI (66.34%, 36.48%, and 35.34%, respectively). The contribution rate of principal componen III was 16.29%, which was mainly correlated with the agricultural output and NDVI (65.65% and 50.88%, respectively). The contribution rate of principal component IV was 9.01%, which was mainly correlated with the degree of land use (89.27%).

In 2010, principal components II, II, and IV all reflect socio-economic background factors. The contribution rate of principal component I was 38.66%, which was mainly correlated with the grassland area and hours of sunshine (70.17% and 36.40%, respectively). The contribution rate of

principal component II was 23.65%, which was mainly correlated with the grassland area, degree of land use, and NDVI, which were 61.74%, 40.41%, and 33.48%, respectively. The contribution rate of principal componen III was 15.11%, which was mainly correlated with the agricultural output and NDVI (62.33% and 51.04%, respectively). The contribution rate of principal component IV was 9.11%, which was mainly correlated with the degree of land use (87.83%).

In 2015, principal components III and IV reflected the socio-economic background factors. The contribution rate of principal component I was 37.61%, which was mainly correlated with the grassland area and hours of sunshine (67.27% and 42.88%, respectively). The contribution rate of principal component II was 24.64%, which was mainly correlated with the grassland area and NDVI (65.00% and 45.41%, respectively). The contribution rate of principal component III was 14.89%, which was mainly correlated with the NDVI and agricultural output (53.00% and 50.47%, respectively). The contribution rate of principal component IV was 9.54%, which was mainly correlated with degree of land use (88.07%).

In Figure 6, in principal component I (PC1), there was a positive correlation between the EVI and grassland area and hours of sunshine from 1990 to 2015. In principal component II (PC2), the EVI was highly correlated with the grassland area and degree of land use from 1990 to 2015. From 2005 to 2015, the EVI was also highly relevant to the agricultural output and average annual precipitation. In principal component III (PC3), there was a high correlation between the EVI and NDVI and agricultural output from 1990 to 2015, as well as a high correlation with the GDP in 1990 and 2015, degree of land use in 1995 and 2000, and industrial output, respectively, in 2005 and 2010. In principal component IV (PC4), from 1990 to 2015, the degree of land use had more than 80% correlation with the EVI.

We can see that principal component I, principal component II, principal component III, and principal component IV reflect the social and economic factors' impact on the ecological conditions. At the same time, principal component I and principal component II reflect the influence of climatic factors, such as the hours of sunshine and average annual precipitation, on the conditions of ecological vulnerability.

## 4. Discussion

### 4.1. Spatial-Temporal Characteristics of Ecological Vulnerability

This study visualized the temporal and spatial changes of ecological vulnerability in Ningxia by using the Tupu and Getis-Ord Gi* spatial analysis models. Tupu clearly shows the transformation of grades of ecological vulnerability in Ningxia. It can be seen from the Tupu analysis that the transformation of the ecological vulnerability grade showed obvious differences in space-time. At the same time, the Getis-Ord Gi* spatial model was used to gather similar ecological vulnerability indexes in spatial statistics, and the results were divided into cold spots, hot spots, and not significant spots.

In the north of Ningxia, the EVI was stable from 1990 to 2015, when the main grade of ecological vulnerability was extreme vulnerability. The north of the study area was also the hot-spot region of the EVI during the study period. The north of Ningxia is mainly constituted of Shizuishan City and Yinchuan City. These two cities feature high-density populations and economic development, which explains why ecological vulnerability was at a high level.

In the west of Ningxia, the EVI was unstable in the study period, with the main grades of ecological vulnerability medium vulnerability and heavy vulnerability. The west of the study area, which is crossed by the Yellow River of China, constitutes Zhongwei City. The region is an important base of grain suppliers, aquatic products, and vegetable facilities in China. The stable economic structure is also the main factor that causes the stabilization of the EVI in Zhongwei City.

In the east of Ningxia, the EVI decreased in the former period but increased later. Based on the analysis spatial heterogeneity, the EVI is not stable in the east of Ningxia. Wuzhong City is the main district in the east of Ningxia. Figure 2 shows that the spatial heterogeneity of the EVI in Wuzhong

is significant. As a node city in the New Silk Road Economic Belt of China, Wuzhong City is active economically, which has a great impact on the local ecological conditions.

In the south of Ningxia, the EVI increased but reversed the trend later. Cold spots are mainly distributed in the south of study area. In addition, and the EVI changed insignificant in this region. Guyuan City in the south of Ningxia has a poor preference in economic and population density, so ecological conditions of this region have rarely been influenced by local human activities.

From the perspective of arid and semi-arid climate zones, the ecological vulnerability in the north of Ningxia, which belongs to an arid region, is much worse than that in central and southern Ningxia, which belongs to a semi-arid region. At the same time, the spatial heterogeneity of the EVI in arid regions is insignificant, while in semi-arid regions it is significant.

### 4.2. The Impact of Ecological Projects on Ecological Vulnerability

Grassland is one of the main driving factors of ecological vulnerability in the study area. The change of the grassland area is influenced by the policy. It can be seen that policies can indirectly affect ecological vulnerability. Due to the ecological characteristics of each patch being different, the implementation of policy is different in space. The Grain for Green Program (GGP), China's ecosystem restoration project, aims to transform farmland, on steep hillsides or in areas of severe desertification, into forestland or grassland in order to increase vegetation coverage and reduce water loss and soil erosion. GGP has a great influence on ecological vulnerability. Ningxia started the GGP in 1999 and the first phase ended in 2006. Therefore, the degree of ecological vulnerability was low in 2005, when the area proportion of slight vulnerability was 51%; after 2005, with a decrease in the intensity of the GGP, the proportion of area with slight vulnerability decreased to 41% by 2010 and sharply decreased to 19% in 2015. It can be seen that GGP has a great impact on the ecological vulnerability of Ningxia. The implementation of GGP in the early stages indeed increased the grassland area in Ningxia, while the economic input was reduced in the latter stages. Additionally, the work of grassland management and grassland maintenance was not very good [37], so the area proportion of slight vulnerability reduced.

Another ecological project that affects grassland area is the Prohibited Grazing Policy (PGP). The PGP is an important ecological service project launched by the Chinese government in 2003 to restore degraded grassland in severely degraded grasslands and ecologically vulnerable areas. Many studies have found that the implementation of the PGP plays an important role in ecological protection, and it has achieved remarkable ecological benefits [25,38,39]. Yanchi County, located in the east of the study area, is a study hot spot of the PGP. According to the results of this study, the ecological vulnerability of Yanchi County in 2005 and 2010 was better than that of other years. The grade of ecological vulnerability was mainly slight vulnerability and light vulnerability in 2005 and 2010, and the degree of ecological vulnerability increased in 2015. The results were similar to other studies [40,41]. It can be seen that the PGP has had a great impact on Yanchi County's environment conditions.

### 4.3. Suggestions

Ningxia is located in the transition zone between the Loess Plateau and the Inner Mongolia Plateau. The Maowusu Sandy Land is in the east and the Tengger Desert is in the west in the north of the study area. Therefore, the ecological situation in the north of the study area is weak. It is easy to cause desertification in the north of the study area if it is not well managed. Desertification is highlighted in Goal 15 of the UN Sustainable Development Goals (SDGs). In this study, based on the study results of ecological vulnerability in Ningxia, we provide reasonable suggestions for sustainable development. The following section gives detailed suggestions on ecological vulnerability protection.

First, the SDGs emphasize the balanced development of society and ecosystems. This study showed that the north of Ningxia, which belongs to an arid region, is mainly at extreme vulnerability in the grades of ecological vulnerability. Meanwhile, the north of Ningxia is mainly the hot spot area of the ENI. The Yellow River is throughout the north of Ningxia, making it have the highest level of economic development in Ningxia. Additionally, it is the region with the fastest development

of urbanization in Ningxia, which leads to changes of land use. Therefore, it is suggested that the government should carry out urban planning reasonably and reduce changes of land use as far as possible, which means the social activities of encroachment on to grassland, forest, and other natural ecosystems should be prohibited in this region. In this way, the unreasonable occupation of the natural ecosystem by social development can be effectively alleviated, and their coordinated development can be guaranteed to a certain extent, so as to promote the sustainable development of society and the ecosystem. Second, the SDGs also focus on desertification. According to the analysis results of the driving factors of ecological vulnerability, the grassland area is the main influencing factor of ecological vulnerability in Ningxia. Meanwhile, changes in grassland areas is closely related to ecological projects. Therefore, it is suggested that the construction of ecological projects should be strengthened in the future, and the management of ecological projects should have strengthened management in the latter stages, so as to ensure long-term sustainable development. In this way, the conditions of ecological vulnerability can be effectively protected, and environmental problems such as desertification can be effectively slowed down in the study area, so as to promote sustainable development. At the same time, the government should increase publicity efforts to raise public awareness of environmental protection, and provide convenient communication platforms for the public, such as Internet platforms, to improve public participation. Finally, according to the spatial heterogeneity of the EVI in Ningxia, we suggest that different levels of government should strengthen inter-regional cooperation and promote the establishment of a unified and synchronized platform for information sharing and monitoring, which could make the ecological condition develop in a better direction in Ningxia.

## 5. Conclusions

In this study, spatial distribution of ecological vulnerability of the Ningxia Hui Autonomous Region in an arid and semi-arid region was studied by using Tupu and Getis-Ord Gi*. Tupu can be used to understand the transformation of the grade of ecological vulnerability in Ningxia. Getis-Ord Gi* can be used to intuitively understand the spatial aggregation state of the EVI in Ningxia. The results of Tupu and cold-hotspot show that the ecological vulnerability of Ningxia has obvious heterogeneity in space. The results showed that conditions of ecological vulnerability were unstable from 1990 to 2015 in the study area, with the degree of ecological vulnerability mainly decreasing from 1990 to 2005 and increasing from 2005 to 2015. During the study period, the degree of ecological vulnerability gradually increased from the south to the north. The hot-spot areas of the EVI were mainly concentrated in the north of Ningxia, while the cold-spots areas of the EVI were scattered in the central and southern areas of Ningxia. At the same time, the results showed that the grassland area was the main driving factor of ecological vulnerability of Ningxia from 1990 to 2015. Through the discussion, it was found that the change in the grassland area is obviously related to the implementation of ecological projects.

The spatial distribution of the EVI was shaped by interaction between human activities and environmental factors. In this study, we found that the ecological vulnerability was greatly affected by ecological projects in Ningxia. Therefore, the government should continue to implement relevant ecological projects in the future and limit construction activities that could cause an increase in the EVI in order to achieve the sustainable development goals in arid and semi-arid areas.

**Author Contributions:** Conceptualization, R.L., R.H., and L.G.; methodology, R.L.; R.H., and Q.Y.; software, R.L. and L.G.; formal analysis, R.L. and R.H.; investigation, R.L. and S.Q.; resources, R.L. and L.G.; data curation, L.G.; writing—original draft preparation, R.L.; writing—review and editing, R.H. and L.G.; project administration, L.G.; funding acquisition, L.G. All authors have read and agreed to the published version of the manuscript.

**Funding:** This research study was supported by the key research project of the Chinese Ministry of Science and Technology (2017YFC0505601) and is an innovation team project of the Chinese Nationalities Affairs Commission, grant number 10301-0190040129.

**Acknowledgments:** We are grateful for the comments of the anonymous reviewers, which greatly improved the quality of this paper.

**Conflicts of Interest:** No conflict of interest exits in this manuscript, and the manuscript has been approved by all authors for publication. The authors declare that the work described was original research that has not been published previously and is not under consideration for publication elsewhere, in whole or in part.

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
