# Peer review of "Spatial Heterogeneous of Ecological Vulnerability in Arid and Semi-Arid Area: A Case of the Ningxia Hui Autonomous Region, China"

_sustainability, doi:10.3390/su12114401_

Round 1

Reviewer 1 Report

The article is a typical case study paper on Ningxia Hui Autonomous on the Loess plateau of China (arid and semi-arid area). The Authors investigated spatial heterogeneous of EVI every 5 years between 1990 and 2015.

My coments are as follow:

  • methods description needs improvement, especially the PCA part. Authors should explain how they divide the factors between each principal component. Moreover the components should have some logical names, not just numbers. Please explain what  makes up the each main (I-IV) components. Also in part with the results Authors should be more precise in PCA part. As an example you can use https://www.mdpi.com/2072-4292/11/20/2359/htm
  • Try to avoid terms like "the regions with high ecological vulnerability are mainly concentrated in the regions ..." (rows 21-22). The term region have a lot of meanings but generaly regions in the regions sounds a little bit strange.
  • authors mentioned that the case study region is the largest Muslims inhabited area of the Hui nationality in China (row 77) - is it a key information from your topic point of view?
  • figure 1 - use sans serif fonts on the maps, in the north-oriented maps do not use the north arrow, on the bottom left map some numbers were marked, but there is no explanation of its meaning on in a legend (e.g. 5 - number of a county),
  • figures 3, 4, 5 - use sans serif fonts on the maps, in the north-oriented maps do not use the north arrow.

Author Response

Manuscript ID: sustainability-799362

Point by point response to the manuscript titled

Spatial heterogeneous of ecological vulnerability in arid and semi-arid area: A case of the Ningxia Hui Autonomous Region, China

Dear Editor,

Thank you very much for your time and comment on our manuscript. On behalf of the co-authors, I am submitting the revised manuscript along with response file. Below are the point-by-point responses to the comments made by reviewers.

Our responses below start with “Response:” and are highlighted in red. The line numbers used here refer to the ones in the Word document (with Track Changes turned to “All Markup”). All line numbers are based on the revised manuscript unless stated otherwise (e.g., “in the previous submission”).

Reviewer #1

  1. Methods description needs improvement, especially the PCA part. Authors should explain how they divide the factors between each principal component. Moreover the components should have some logical names, not just numbers. Please explain what makes up the each main (I-IV) components. Also in part with the results Authors should be more precise in PCA part. As an example you can use https://www.mdpi.com/2072-4292/11/20/2359/htm

Response: Thanks for reviewer’s comments. We have revised this section according to the reviewer’s comment.

First, we have asked professionals to modify the language used to describe the methods section.

Second, we have added the PCA method and contribution of original indexes of ecological vulnerability to principal components in method section, and the specific content is as follows:

Method (see lines 125-139)

“…SPCA is to add spatial features on the basis of PCA, and its calculation principle is consistent with PCA [1, 15,16].

PCA is a statistical analysis method that transforms multiple variables into a few principal components (composite indicators) through dimensionality reduction [32-34]. In this study, PCA is to make a linear combination of 12 standardized indexes to make them become new comprehensive indexes. The correlation coefficient matrix is solved to obtain the eigenvectors, thus obtaining 12 principal component results. The number of principal components was determined by the standard of cumulative contribution rate ≥ 85%. We get our final principal component result. The calculation formula is as follows:

(1)

(2)

(3)

(4)

Where R is correlation coefficient matrix; Z is the standardized value of each selected index; n is the number of indexes; λ is the eigenvalues of the R correlation coefficient matrix; I is the identity matrix; CCR is the cumulative contribution rate ; m is The number of principal components was determined; P is the matrix containing values of every considered principal component; W is m number of eigenvectors with the largest eigenvalues were selected to form dimensional matrix.

The SPCA was obtained by calculating PCA in ArcGIS 10.4…”

The table of Contribution of original determinants of ecological vulnerability to principal components too big to be added in the manuscript.

Finally, we added some detailed explanations of each main (I-IV) components in the driving factors analysis section (3.4) of the results section. See lines 243-306.

Table. Contribution of original determinants of ecological vulnerability to principal components.

Principal Component

Average annual precipitation

DEM

Average annual temperature

Hours of sunshine

Population density

Agricultural output

NDVI

Degree of land use

Industrial output

GDP

Grassland area

Soil erosion

1990a

-0.4705

-0.2089

0.1759

0.4153

0.0209

0.1968

-0.0642

-0.2218

0.1246

0.1686

0.6287

-0.0045

0.3308

0.1715

-0.1555

-0.3086

0.0126

-0.0141

0.1628

0.4213

-0.0246

-0.0233

0.7323

0.0052

-0.0409

-0.0689

0.0150

0.0042

0.1054

0.4769

0.6068

0.2764

0.3015

0.4114

-0.2238

0.0053

-0.2385

-0.1432

0.1208

0.2323

-0.0203

-0.1212

-0.3755

0.8253

0.0251

0.0030

-0.1276

-0.0060

1995a

-0.3904

-0.1787

0.1165

0.2948

0.0224

0.2128

-0.0066

-0.1592

0.1112

0.0283

0.7942

-0.0036

0.4622

0.2422

-0.1608

-0.3290

-0.0359

-0.2978

-0.0880

0.3340

-0.1586

-0.0396

0.5980

0.0033

-0.0761

-0.0159

-0.0152

0.0278

0.1138

0.3983

0.4357

0.7691

0.1907

0.0529

-0.0297

0.0065

0.2839

0.2555

-0.2014

-0.1928

0.1028

0.3647

0.5975

-0.5147

0.0427

0.0191

0.0927

0.0119

2000a

-0.3913

-0.2080

0.1440

0.3758

0.0247

0.3490

-0.0460

-0.2149

0.1329

0.0434

0.6719

-0.0057

0.3668

0.1926

-0.1576

-0.3214

0.0079

-0.0796

0.2561

0.3990

-0.0477

-0.0102

0.6832

0.0074

0.0262

-0.0679

0.0059

-0.0366

0.0680

0.6826

0.6002

0.2217

0.1841

0.0710

-0.2726

0.0069

-0.2626

-0.1710

0.1292

0.2041

-0.0078

-0.0543

-0.3034

0.8601

0.0519

0.0100

-0.0759

-0.0078

2005a

-0.3781

-0.1969

0.1426

0.4209

0.0363

0.2295

-0.1535

-0.2269

0.2215

0.0508

0.6634

-0.0058

0.3303

0.1618

-0.1229

-0.3724

0.0378

0.1178

0.3534

0.3648

0.0008

0.0146

0.6614

0.0071

-0.0275

-0.1413

0.0864

0.0142

0.1373

0.6565

0.5088

0.0100

0.3792

0.0923

-0.3323

0.0037

-0.1869

-0.1343

0.0862

0.2023

0.0098

-0.0148

-0.2565

0.8927

0.1577

0.0343

-0.0980

-0.0051

2010a

-0.3412

-0.1883

0.1401

0.3640

0.0436

0.2508

-0.1244

-0.2099

0.2733

0.0676

0.7017

-0.0050

0.3232

0.1861

-0.1676

-0.4046

0.0147

0.0956

0.3348

0.4041

-0.0560

-0.0004

0.6174

0.0069

-0.0402

-0.1363

0.0775

-0.0486

0.1031

0.6233

0.5104

0.0783

0.4223

0.1164

-0.3374

0.0033

-0.1833

-0.1406

0.0904

0.2198

0.0059

-0.0576

-0.2939

0.8783

0.1421

0.0369

-0.0869

-0.0052

2015a

-0.3167

-0.1815

0.1210

0.4288

0.0283

0.1647

-0.2466

-0.2474

0.2535

0.0393

0.6727

-0.0060

0.3011

0.1167

-0.0752

-0.3472

0.0237

0.1709

0.4541

0.3326

0.0088

0.0130

0.6500

0.0073

0.0474

-0.2219

0.1780

0.1261

0.0663

0.5047

0.5300

-0.1625

0.4737

0.0700

-0.3243

0.0022

-0.0850

-0.1432

0.0893

0.2207

0.0247

0.0860

-0.2420

0.8807

0.2421

0.0373

-0.1158

-0.0047

  1. Try to avoid terms like "the regions with high ecological vulnerability are mainly concentrated in the regions ..." (rows 21-22). The term region have a lot of meanings but generaly regions in the regions sounds a little bit strange.

Response:  Thanks for the reviewer's comment. The original sentence was changed to "the regions with high ecological vulnerability are mainly concentrated in the north of the study area with the high growth of economic …" (see lines 22-24)

  1. authors mentioned that the case study region is the largest Muslims inhabited area of the Hui nationality in China (row 77) - is it a key information from your topic point of view?

Response: Thanks for the reviewer's comment. It is only a fact description that the case study region is the largest Muslims inhabited area of the Hui nationality in China, which is not necessarily related to the study. Therefore, it has been deleted according to the reviewer's comment (see lines 81-82).

  1. figure 1 - use sans serif fonts on the maps, in the north-oriented maps do not use the north arrow, on the bottom left map some numbers were marked, but there is no explanation of its meaning on in a legend (e.g. 5 - number of a county),

Response: Thanks to the reviewer's comments. According to the reviewer's comments, we have made the following modifications:

Frist, we modified the font in the figure into a sans-serif font.

Second, we delete the north arrow from the diagram.

Finally, we have a detailed explanation of the Number meanings in the figure in the notes. The modification is as follows (see Figure 1 in lines 100-105):

  1. figures 3, 4, 5 - use sans serif fonts on the maps, in the north-oriented maps do not use the north arrow.

Response: Thanks to the reviewer's comments. According to the reviewer's comments, we modified the font in the figure into a sans-serif font, and delete the north arrow in Figures. (see Figure 3 in line 197, Figure 4 in line 218, Figure 5 in line 239).

Reviewer 2 Report

The paper reports result from the elaboration of geo-information data to calculate the ecological vulnerability in an arid/semiarid region in a Northern region of China

GENERAL COMMENTS

The article approaches an interesting and up-to-date subject dealing with the long term sustainability of land exploitation regarding, mainly, the use of the soil.

Being interesting, the manuscript seems more an exercise of the use of geographical and geostatistical instruments and methods without a real enhancement of knowledge on the subject. The suggestions from the results of the study are vague and generical and can be advised without any scientific study on the background.

I suggest the authors to introduce their manuscript the UN Sustainable Development Goals

https://www.un.org/sustainabledevelopment/sustainable-development-goals/

in particular, how the results/evidence can contribute to support progress to achieving some of such goals (L328). This can increase the interest of the potential audience and, potentially, impact policy development (L292-298).

SPECIFIC COMMENTS

L44 I would change “…. it is very important to assess the ecological vulnerability..” in something less general and more actual such as“ ..to assess ecological vulnerability is relevant because ….”

L49 “… and so on.” is colloquial. It is not acceptable in a scientific manuscript. Replace with something like “ … and other (less popular?) techniques.”

L55-66 The authors make a list of factors affecting ecological vulnerability without citing any aspect related to agriculture while later suggesting enhancement  of agriculture and grazing techniques as one of the methods to reduce the ecological vulnerability of the area (see L261-262+L292-298+L317)

L77 I would not make reference to Muslim religion of the population. We are in a scientific paper not anthropological or sociological journal unless the “Religion” is one of the factors taken into account by the statistic

L108-109 “1x1 km” or “ 1 km2”. As it is written now is not correct

L116 Soil degradation, in particular soil erosion, is one of the major threat to face globally, and having a large impact on agricultural lands and, thus, on ecological vulnerability. I suggest the authors to further consider this subject in the manuscript with references to specific study on it. Here are some examples:

  • Biddoccu1 M. et Al. 2017. Assessment of long-term soil erosion in a mountain vineyard, Aosta valley (NW Italy). Land Degradation and Development, 29(3), DOI:10.1002/ldr.2657
  • Bagagiolo G. et Al. Effects of rows arrangement, soil management, and rainfall characteristics on water and soil losses in Italian sloping vineyards. Environmental Research, 2018, 166, 690-704

L281       Very generic suggestions that are not based on the outcome of the study. Make the effort to relate the measures proposed to the specific results of the study.

L318-319 Avoid referring to components of the PCA. We are in Conclusion, where a summary of the results and their impact have to be discussed. Results from SPCA have been already explained in previous Results sections.

Author Response

Manuscript ID: sustainability-799362

Point by point response to the manuscript titled

Spatial heterogeneous of ecological vulnerability in arid and semi-arid area: A case of the Ningxia Hui Autonomous Region, China

Dear Editor,

Thank you very much for your time and comment on our manuscript. On behalf of the co-authors, I am submitting the revised manuscript along with response file. Below are the point-by-point responses to the comments made by reviewers.

Our responses below start with “Response:” and are highlighted in red. The line numbers used here refer to the ones in the Word document (with Track Changes turned to “All Markup”). All line numbers are based on the revised manuscript unless stated otherwise (e.g., “in the previous submission”).

Reviewer#2

GENERAL COMMENTS

The article approaches an interesting and up-to-date subject dealing with the long-term sustainability of land exploitation regarding, mainly, the use of the soil.

Response: We thank the reviewer for his or her valuable time and the constructive comments on this manuscript. We have revised the manuscript based on the reviewer’s following suggestions.

Being interesting, the manuscript seems more an exercise of the use of geographical and geostatistical instruments and methods without a real enhancement of knowledge on the subject. The suggestions from the results of the study are vague and generical and can be advised without any scientific study on the background.

Response: Thanks for the reviewer's comments.

First, based on the first question, we add content in the following aspects:

(1) Result (see lines183-185, 199-201, 220-224)

“The spatial change of ecological vulnerability is shown in Figure 3. The spatial statistical model is used for data visualization, which shows the spatial and temporal variation of ecological vulnerability in Ningxia more clearly and intuitively.” (see lines183-185).

“Tupu is a spatial model that visualizes the transformation between different grades of ecological vulnerability at each patch. Tupu analysis can more clearly show the dynamic change of ecological vulnerability in Ningxia from 1990 to 2015(Figure 4.)” (see lines199-201).

“Cold-hotspot is a spatial model used to display spatial aggregation calculated by Getis-Ord Gi*. The calculation of the EVI of Ningxia from 1990 to 2015 based on the Cold-hotspot can intuitively see spatial aggregation of similar values. At the same time, by comparing the spatial distribution of cold spots and hot spots in different years, we can get the differences in the spatial change of EVI on the time scale (Figure 5).” (see lines220-224).

(2) Discussion (see lines 328-334, 360-362).

“This study visualized the temporal and spatial changes of ecological vulnerability in Ningxia by using Tupu and Getis-Ord Gi * spatial analysis models. Tupu clearly shows the transformation of grades of ecological vulnerability in Ningxia. It can be seen from the Tupu analysis that the transformation of the ecological vulnerability grade showed obviously different in space-time. At the same time, Getis-Ord Gi* spatial model was used to gather the similar ecological vulnerability indexes in spatial statistics, and the results were divided into cold spots, hot spots and not significant spots.” (see lines 328-334).

“It can be seen that policies can indirectly affect ecological vulnerability. Due to the ecological characteristics of each patch is different, the implementation of the policy is different in space.” (see lines 360-362).

(3) Conclusion (see lines 444-448).

“The Tupu can be used to understand the transformation of ecological vulnerability grade in Ningxia. Getis-Ord Gi* can be used to intuitively understand the spatial aggregation state of EVI in Ningxia. The results of Tupu and Cold-hotspot show that the ecological vulnerability of Ningxia has obvious heterogeneity in space.”.

Second, we made some modifications to the Discussion section (4.3), and each suggestion was based on the research results (see lines 394-414). The modification is as follows:

Discussion (see lines 394-414)

“First, according to the spatial-temporal results of the ecological vulnerability in the study area, the north of Ningxia, which belongs to the arid region, is mainly at extreme vulnerability in the grades of ecological vulnerability. Meanwhile, in the Cold-hot spot results, the north of Ningxia is mainly the hotspot area of ENI. At the same time, the Yellow River is throughout the north of Ningxia, making it become the highest level of economic development in Ningxia. Also, it is the region with the fastest development of urbanization in Ningxia, which leads to the change of land use. Therefore, it is suggested that the government of this region should carry out urban planning reasonably and reduce the change of land use as far as possible, which means is the social activities of encroachment on grassland, forest and other natural ecosystems should be prohibited in this region. Second, according to the analysis results of driving factors of ecological vulnerability, grassland area is the main influencing factor of ecological vulnerability in Ningxia. Meanwhile, the change of grassland area is closely related to ecological projects. Therefore, it is suggested that the construction of ecological projects should be strengthened in the future, and the management of ecological projects should be strengthened management in the later stage, so as to ensure the long-term sustainable development. At the same time, the government should increase publicity efforts to raise public awareness of environmental protection, and provide convenient communication platforms for the public, such as Internet platforms, to improve public participation. Finally, according to the spatial heterogeneity of EVI in Ningxia, we suggest that different levels of governments should strengthen inter-regional cooperation and promote the establishment of a unified and synchronized platform for information sharing and monitoring, which can make the ecological condition develop in a better direction in Ningxia.”.

I suggest the authors to introduce their manuscript the UN Sustainable Development Goals

https://www.un.org/sustainabledevelopment/sustainable-development-goals/

in particular, how the results/evidence can contribute to support progress to achieving some of such goals (L328). This can increase the interest of the potential audience and, potentially, impact policy development (L292-298).

Response: Thanks for the reviewer's comments. We have introduced the UN Sustainable Development Goals in the manuscript (see lines 386-390). The add is as follows:

    “Ningxia is located in the transition zone between the Loess plateau and the Inner Mongolia plateau. Maowusu sandy land in the east and Tengger desert in the west in the north of the study area. Therefore, the ecological situation in the north of the study area is week. It is easy to cause desertification in the north of the study area if it is not well managed. Desertification is highlighted in Goal 15 of the UN Sustainable Development Goals (SDGs).”

SPECIFIC COMMENTS

  1. L44 I would change “…. it is very important to assess the ecological vulnerability…” in something less general and more actual such as “..to assess ecological vulnerability is relevant because ….”

Response: Thanks for the reviewer's comments. We have modified this sentence according to the reviewer's opinion.

“…. it is very important to assess the ecological vulnerability..” change to “The sustainable development of arid and semi-arid regions is related to assess ecological vulnerability, because the arid and semi-arid areas are especially sensitive to the variable environmental changes and human influences.” (see lines 44-45).

  1. L49 “… and so on.” is colloquial. It is not acceptable in a scientific manuscript. Replace with something like “… and other (less popular?) techniques.”

Response: Thanks for the reviewer's comments. We have made modifications according to the reviewer's opinion.

“… and so on.” change to “…and other techniques.”(see lines 52-53).

  1. L55-66 The authors make a list of factors affecting ecological vulnerability without citing any aspect related to agriculture while later suggesting enhancement of agriculture and grazing techniques as one of the methods to reduce the ecological vulnerability of the area (see L261-262+L292-298+L317)

Response: Thanks for the reviewer's comment. We have deleted the content about enhancement of agriculture and grazing techniques from the suggested section. See lines 423-431.

  1. L77 I would not make reference to Muslim religion of the population. We are in a scientific paper not anthropological or sociological journal unless the “Religion” is one of the factors taken into account by the statistic

Response: Thanks for the reviewer's comments. It is only a fact description that the case study region is the largest Muslims inhabited area of the Hui nationality in China, which is not necessarily related to the study. Therefore, it has been deleted according to the reviewer's comment (see lines 81-82).

  1. L108-109 “1x1 km” or “1 km2”. As it is written now is not correct

Response: Thanks for the reviewer's comment. We have made modifications based on articles published in the Sustainability. See lines 114-115.

“1x1 km2” change to “1x1 km”.

  1. L116 Soil degradation, in particular soil erosion, is one of the major threat to face globally, and having a large impact on agricultural lands and, thus, on ecological vulnerability. I suggest the authors to further consider this subject in the manuscript with references to specific study on it. Here are some examples:

Biddoccu1 M. et Al. 2017. Assessment of long-term soil erosion in a mountain vineyard, Aosta valley (NW Italy). Land Degradation and Development, 29(3), DOI:10.1002/ldr.2657

Bagagiolo G. et Al. Effects of rows arrangement, soil management, and rainfall characteristics on water and soil losses in Italian sloping vineyards. Environmental Research, 2018, 166, 690-704

Response: According to the comments of reviewers, we will make some explanations about why there is no explanation in the Result and Discussion section about the impact of soil erosion on the ecological fragility of the study area:

First, we can be sure that soil erosion will affect ecological vulnerability. Therefore, we chose soil erosion as one of our indicators to assess ecological vulnerability of Ningxia.

Second, according to the calculation results of PCA, the contribution of soil erosion to all principal components is very small, which is basically less than 1% (Table 2). Therefore, in the results section, we did not conduct in-depth research on soil erosion.

Finally, other studies show that soil erosion is a minor problem in Ningxia's ecological problems, while grassland degradation and desertification are the main problems in Ningxia. The references are as follows:

Du, H., Xue, X., & Wang, T. (2015). Mapping the risk of water erosion in the watershed of the Ningxia-Inner Mongolia reach of the Yellow River, China. Journal of Mountain Science, 12(1), 70–84. doi:10.1007/s11629-013-2861-8

Du, H., Dou, S., Deng, X., Xue, X., & Wang, T. (2016). Assessment of wind and water erosion risk in the watershed of the Ningxia-Inner Mongolia Reach of the Yellow River, China. Ecological Indicators, 67, 117–131. doi:10.1016/j.ecolind.2016.02.042

  1. L281 Very generic suggestions that are not based on the outcome of the study. Make the effort to relate the measures proposed to the specific results of the study.

Response: Thanks for the reviewer's comment. we made some modifications to the Discussion section (4.3), and each suggestion was based on the research results (see lines 394-414.

  1. L318-319 Avoid referring to components of the PCA. We are in Conclusion, where a summary of the results and their impact have to be discussed. Results from SPCA have been already explained in previous Results sections.

Response: Thanks to the reviewer's comments, we have deleted the content of PCA in the result part (see lines 457-461) and added it as follows:

“At the same time, the results showed that the grassland area was mainly driving factor of ecological vulnerability of Ningxia from 1990 to 2015. Through the discussion, it is found that the change of grassland area is obviously related to the implementation of ecological project.” (see lines 453-455).

Round 2

Reviewer 1 Report

All my comments were considered. The article is much better after the revision. In my opinion it can be publish in present form.

Author Response

Manuscript ID: sustainability-799362

Point by point response to the manuscript titled

Spatial heterogeneous of ecological vulnerability in arid and semi-arid area: A case of the Ningxia Hui Autonomous Region, China

Dear Editor,

Thank you very much for your time and comment on our manuscript. On behalf of the co-authors, I am submitting the revised manuscript along with response file. Below are the point-by-point responses to the comments made by reviewers.

Our responses below start with “Response:” and are highlighted in red. The line numbers used here refer to the ones in the Word document (with Track Changes turned to “All Markup”). All line numbers are based on the revised manuscript unless stated otherwise (e.g., “in the previous submission”).

Reviewer#1

All my comments were considered. The article is much better after the revision. In my opinion it can be publish in present form.

Response: Thanks for the reviewer's approval of our revised manuscript.

Reviewer 2 Report

The authors adreesed most of the comments and suggestion raised on the previosu version of the manuscript

Nevertheless some typo/mistakes are present

L388 weak instead of week

I recommended the authors to introduce their investigation on the lights of the issues the UN Sustainable Development Goals deal with and how the results of their study could contribute to achieve them.

They just introduce the subject in L390 without any broad comment and without any organic introduction in the text. The benefit of the results of the study for the achievments on the UN Goals shoud be clearer.

Author Response

Manuscript ID: sustainability-799362

Point by point response to the manuscript titled

Spatial heterogeneous of ecological vulnerability in arid and semi-arid area: A case of the Ningxia Hui Autonomous Region, China

Dear Editor,

Thank you very much for your time and comment on our manuscript. On behalf of the co-authors, I am submitting the revised manuscript along with response file. Below are the point-by-point responses to the comments made by reviewers.

Our responses below start with “Response:” and are highlighted in red. The line numbers used here refer to the ones in the Word document (with Track Changes turned to “All Markup”). All line numbers are based on the revised manuscript unless stated otherwise (e.g., “in the previous submission”).

Reviewer#2

The authors adreesed most of the comments and suggestion raised on the previosu version of the manuscript

Nevertheless some typo/mistakes are present

Response: We thank the reviewer for his or her valuable time and the constructive comments on this manuscript. We have revised the manuscript based on the reviewer’s following suggestions.

  1. L388 weak instead of week

Response: Thanks for the reviewer's comments. We have made modifications according to the reviewer's comment. At the same time, we checked the grammar and vocabulary of the manuscript.

“week” change to “weak” (see line 387).

  1. I recommended the authors to introduce their investigation on the lights of the issues the UN Sustainable Development Goals deal with and how the results of their study could contribute to achieve them.

They just introduce the subject in L390 without any broad comment and without any organic introduction in the text. The benefit of the results of the study for the achievements on the UN Goals should be clearer.

Response: Thanks for the reviewer's comments. We have introduced the UN Sustainable Development Goals in the manuscript. We add content as following (see lines393-405, 406-413):

“First, SDGS emphasizes the balanced development of society and ecosystem. This study showed that the north of Ningxia, which belongs to the arid region, is mainly at extreme vulnerability in terms of the grades of ecological vulnerability. Meanwhile, the north of Ningxia is mainly the hotspot area of ENI. The Yellow River is throughout the north of Ningxia, making it become the highest level of economic development in Ningxia. Also, it is the region with the fastest development of urbanization in Ningxia, which leads to the change of land use. Therefore, it is suggested that the government should carry out urban planning reasonably and reduce the change of land use as far as possible, which means the social activities of encroachment on grassland, forest and other natural ecosystems should be prohibited in this region. In this way, the unreasonable occupation of the natural ecosystem by social development can be effectively alleviated, and their coordinated development can be guaranteed to a certain extent, so as to promote the sustainable development of the society and the ecosystem.” (see lines 393-405).

“Second, SDGs also focuses on desertification. According to the analysis results of driving factors of ecological vulnerability, grassland area is the main influencing factor of ecological vulnerability in Ningxia. Meanwhile, the change of grassland area is closely related to ecological projects. Therefore, it is suggested that the construction of ecological projects should be strengthened in the future, and the management of ecological projects should be strengthened management in the later stage, so as to ensure the long-term sustainable development. In this way, the vulnerability ecological conditions can be effectively protected, and environmental problems such as desertification can be effectively slowed down in the study area, so as to promote sustainable development.” (see lines 406-413).
